# Real-World Efficacy of Midostaurin in Aggressive Systemic Mastocytosis

**DOI:** 10.3390/jcm10051109

**Published:** 2021-03-07

**Authors:** Aneta Szudy-Szczyrek, Oliwia Bachanek-Mitura, Tomasz Gromek, Karolina Chromik, Andrzej Mital, Michał Szczyrek, Witold Krupski, Justyna Szumiło, Zuzanna Kanduła, Grzegorz Helbig, Marek Hus

**Affiliations:** 1Chair and Department of Haematooncology and Bone Marrow Transplantation, Medical University of Lublin Staszica Street 11, 20-081 Lublin, Poland; oliwiabachanekmitura@umlub.pl (O.B.-M.); togromek@gmail.com (T.G.); 2Department of Hematology and Bone Marrow Transplantation, Medical University of Silesia in Katowice, 40-032 Katowice, Poland; karolina.torbaa@gmail.com (K.C.); ghelbig@o2.pl (G.H.); 3Department of Hematology and Transplantology, Medical University of Gdańsk, 80-211 Gdańsk, Poland; amital@wp.pl; 4Chair and Department of Pneumonology, Oncology and Allergology, Medical University of Lublin, 20-090 Lublin, Poland; michal.szczyrek@umlub.pl; 5II Department of Medical Radiology, Medical University of Lublin, 20-081 Lublin, Poland; witold.krupski@umlub.pl; 6Chair and Department of Clinical Pathomorphology, Medical University of Lublin, 20-090 Lublin, Poland; justyna.szumilo@umlub.pl; 7Department of Hematology and Bone Marrow Transplantation, University of Medical Sciences in Poznan, 61-001 Poznań, Poland; zuzanna.kandula@skpp.edu.pl

**Keywords:** aggressive systemic mastocytosis, midostaurin, mast cells, tryptase, *KIT D816V* mutation

## Abstract

In April 2017 midostaurin was approved by the US Food and Drug Administration for the treatment of patients with aggressive systemic mastocytosis (ASM). So far, very limited real world data on its efficacy is available. Thirteen patients aged from 48 to 79 years, who received midostaurin in the early access program, were included in the study. Midostaurin was used both in first (*n* = 5) and subsequent lines of treatment (*n* = 8). The median duration of exposure was 9 months. Most patients (77%, *n* = 10) had a clinical improvement already as soon as the second month of therapy. Objective response was noted in 4 (50%) of eight evaluated patients. Among responders, we observed a decrease in serum tryptase level (median 74.14%) and bone marrow infiltration by mast cells (median 50%) in the sixth month of treatment. In one case, in the 10th month of treatment, allogenic stem cell transplantation was performed, achieving complete remission. Five patients died, three due to progression of disease, one in the course of secondary acute myeloid leukemia and one due to reasons not related to mastocytosis. Treatment is ongoing in seven patients. We found that midostaurin therapy is beneficial to patients with ASM.

## 1. Introduction

Aggressive systemic mastocytosis (ASM) according to the 2016 World Health Organization (WHO) classification is one of the advanced forms of systemic mastocytosis (SM), a rare neoplasm of the myeloid lineage characterized by impaired expansion and accumulation of mast cells (MCs) in the bone marrow and other organs—skin, liver, spleen, and lymph nodes. In most of the patients (>90%), a somatic mutation in the *KIT* gene in codon 816, encoding a receptor protein with tyrosine kinase activity is detected. Diagnosis of ASM is associated with poor prognosis, the estimated median survival is 3.5 years. Patients are at risk of leukemic transformation into mast cell leukemia (MCL) or acute myeloid leukemia (AML), with observed rate of progression about 5% [1,2].

The clinical picture of ASM is very diverse. Disease symptoms may result from both the release of mediators from MCs and organ damage associated with infiltration by mast cells. Clinical symptoms resulting from neoplastic infiltration (the so-called “C” findings) include cytopenia, bone lesions, hepatomegaly with impaired liver function and/or portal hypertension, spleen enlargement with hypersplenism, and weight loss due to gastrointestinal involvement. For the diagnosis of ASM it is necessary to observe at least one “C” finding [3,4,5].

Treatment options for patients with ASM are still limited. The mainstay of treatment is cytoreductive therapy and the treatment of symptoms associated with MC mediator. Allogeneic stem cell transplantation (alloSCT) is currently the only curative option [6,7].

Among the new drugs introduced into therapy, the greatest hope is recently raised by midostaurin—a multi-targeted protein kinase inhibitor. In vitro, midostaurin or its active metabolites inhibit both wild-type and *D816V* mutant *KIT* tyrosine kinases [8,9] as well as other kinases, including FLT3 kinase, platelet-derived growth factor α (PDGFR-α) and β (PDGFR-β) receptors, Src protein tyrosine kinase, and vascular endothelial growth factor receptor (VEGFR) [10]. In preclinical studies, it has been shown that the drug inhibits MC proliferation and the release of histamine [11].

In April 2017, the US Food and Drug Administration (FDA) approved midostaurin for the treatment of adult patients with aggressive systemic mastocytosis (ASM), systemic mastocytosis associated with hematological neoplasm (SM-AHN) or mast cell leukemia (MCL), regardless of the *KIT D816V* mutation status. The registration was based on the satisfactory results of clinical trial #CPKC412D2201 [12]. So far, there are no data on the use of the drug in ASM therapy in real world clinical practice.

## 2. Methods

In this study, we analyzed patients diagnosed with ASM treated with midostaurin at three academic centers in Poland: The Department of Hematooncology and Bone Marrow Transplantation in Lublin, Department of Hematology and Bone Marrow Transplantation in Katowice and Department of Hematology and Transplantology in Gdańsk. Patients were followed up since January 2019 to January 2021. Patients received midostaurin via early access program. Data collection and analysis were performed independently of Novartis Pharmaceuticals.

All patients met the criteria for ASM diagnosis according to the WHO classification with at least one “C” finding of organ damage. Diagnosis of mastocytosis was confirmed by a bone marrow biopsy in each case. Bone marrow sections were analyzed by immunohistochemistry using antibodies against CD117 [13]. Flow cytometry was used to detect an atypical immunophenotype of MCs, in particular expression of CD2, CD25, and CD117 [14]. Genetic tests were performed in all patients to assess the *c-KIT* mutation. The material for the study was peripheral blood and bone marrow samples in each case. The Sanger sequencing method was used for the analysis [15]. To evaluate the organ involvement abdominal ultrasound, X-ray, densitometry and whole-body low-dose CT scans were used.

Adult patients with Eastern Cooperative Oncology Group (ECOG) performance status ≤3, with adequate liver and kidney function, were eligible for treatment. Exclusion criteria were QTcF prolongation >450 msec, elevated liver enzymes [aspartate aminotransferase (AST) and alanine aminotransferase (ALT) >2.5 ULN or in the case of mastocytosis-related liver injury × ULN), and elevated bilirubin (>1.5 ULN).

All patients gave informed consent to treatment and participation in the study. The date of last follow-up for the present report was 31 January 2021.

The primary endpoint of the study was the best clinical response according to the modified International Working Group-Myeloproliferative Neoplasms Research and Treatment and European Competence Network on Mastocytosis (IWG-MRT-ECNM) response criteria for patients with ASM, MCL and SM-AHN achieved after six 4-week treatment cycles. The response was assessed in relation to systemic symptoms associated with mediators and to findings “B” and “C” of organ damage [5]. We analyzed the duration of exposure to midostaurin, treatment interruptions and the frequency of selected adverse events (AEs). The severity of AE’s was assessed according to the Common Terminology Criteria for Adverse Events (CTCAE) scale v5.0.

## 3. Results

### 3.1. Study Group

Thirteen patients were enrolled in the study, including eight women and five men. The median age was 63 years (range 48–79).

The percentage of bone marrow MCs ranged from 2 to 80% (median 20%). The MC aggregates were paratrabecular or perivascular in location and associated with fibrosis in most cases (*n* = 9). Fibrosis by histochemical stains corresponded to an MF-3 grade in four, an MF-2 in three and an MF-1 in two case. The morphological demonstration of compact MC infiltrates were detected in the skin in five patients, in the iliac plate in one case and in the axillary lymph node in one case. Aberrant CD2 and CD25 expression on MCs was confirmed by flow cytometry in almost all patients (*n* = 12). The median percentage of MCs with an atypical immunophenotype was 0.57% (range 0.1–28.6%). In eleven patients the mutation in the *KIT D816V* gene was present, and in one patient it was accompanied by the *KIT D816H* mutation. In the patients with concurrent grade ≥2 fibrosis (*n* = 7) *JAK2 V617F* analyses were performed to exclude primary myelofibrosis (PMF).

All patients reported symptoms related to the release of mediators from MCs. Abdominal pain, diarrhea, hypotension, dizziness, and occasional flushing of the skin dominated. Typical skin discoloration, with pathomorphological evidence of MC infiltration, was present in five patients. In one patient with a history of allergic reactions and anaphylactic shock, allergy to Hymenoptera venom was confirmed.

Spleen enlargement was found in nine, hepatomegaly in nine, ascites in three generalized lymphadenopathy in four patients. Varied bone lesions: diffuse small (<0.5 cm) asymptomatic osteolytic lesions, osteosclerosis or generalized osteoporosis were confirmed in almost all patients (*n* = 12). Large osteolyses were reported in eight patients including pathological fracture in three case. Clinically important weight loss was reported in six patients, hypoalbuminemia grade 2 in one case. Hematologic manifestations included anemia grade 2 (*n* = 5), transfusion-dependent anemia (*n* = 2), thrombocytopenia grade 1 (*n* = 2), grade 2 (*n* = 1), grade 3 (*n* = 2) and neutropenia grade 4 (*n* = 2). Patients with “C” findings included eight patients with measurable changes and five patients with only non-measurable changes (ascites, osteolytic changes, weight loss). Patient characteristics are presented in Table 1.

### 3.2. Treatment

Patients received midostaurin 100 mg orally twice daily with meals for 28-day treatment cycles until disease progression or intolerable toxicity. In one case, in a patient with coexisting renal failure with reduced estimated glomerular filtration rate (eGFR) of 40 mL/min, it was decided to reduce the dose to 50 mg twice a day. All patients received prophylaxis of symptoms related to the release of MCs with oral H1 and H2 antagonists (fexofenadine/loratidine, ranitidine/famotidine). There was no need for chronic glucocorticoid use in any of the patients. In eight patients with large osteolytic lesions treatment with bisphosphonates was started. Disodium pamidronate was used iv. in doses adjusted to the eGFR.

In five patients, midostaurin was used in the 1st line, in six patients—in the 2nd, and in two cases—in the 5th line of treatment. Previous treatments included cladribine (2Cd-A), hydroxycarbamide (HU), dasatinib and imatinib in a patient without a confirmed *c-KIT* mutation. In two cases, 2 CD-A was used in the 1st and further lines of treatment. The median time from the initiation of first-line treatment to was 6.5 months (4–62 months).

The median duration of treatment with midostaurin was 9 months (range: 1–21 months). 

### 3.3. Responses and Outcomes

Among thirteen enrolled patients, ten had some clinical benefit (77%), three patients progressed in the 2nd month of therapy and died. Clinical improvement (CI) was observed in all responders, with partial remission regarding symptoms associated with the release of mediators from MCs already present in the 2nd month of treatment. There was relief from gastric ailments, allergic reactions, episodes of hypotension, bone pain and a weight gain with a median of 5 kg (range 3–15 kg). A gradual reduction of skin lesions was observed in a patients with maculopapular rash typical of mastocytosis.

Serum tryptase level was systematically assessed and a bone marrow biopsy was repeated after 6 months of therapy in eight patients. A decrease in tryptase concentration was observed in all of them with median change of 74 ng/mL (range: 36.1–154 ng/mL), which was 74.14% (range: 43.2–85.5%) (Figure 1). Moreover, there was a reduction in bone marrow infiltration by MCs in bone marrow biopsy, with median change of 10% after 6 months of treatment (range 3–73%), which constituted 50% of the baseline involvement (range 30–91.25%) %) (Figure 2). In patients with hepatosplenomegaly during the first 3 months of treatment, a decrease in the initial enlargement by about 25% was observed.

During the first 6 months of treatment, 4 out of 8 (50%) patients with measurable “C” findings achieved response—major remission (MR) (*n* = 1), partial remission (PR) (*n* = 3). In the remaining nine patients, the assessment of the response to treatment according to the IWG-MRT-ECNM criteria was not possible due to the absence of measurable “C” findings (*n* = 5) and deaths (*n* = 4) (Table 2). Among patients with unmeasurable “C” findings, we considered an improvement of general condition, a decrease of serum tryptase concentration (median 54.1 ng/mL; range: 41–129.4 ng/mL) and bone marrow MCs infiltration (median 10%; range: 3–35%) in the 6th month to be a clinical response for midostaurin.

In one case, a 52-year-old man with *KIT D816V+* mutation, advanced bone disease and hepatomegaly, in the 10th month of midostaurin therapy, with CI (general symptoms disappearance, bone marrow infiltration decreased by 30%, serum tryptase level decreased <20 ng/mL), it was decided to carry out alloSCT. In a control bone marrow biopsy 6 months after transplantation we found no MCs infiltration and no expression of *KIT D816V +.*

There were 5 deaths, one in the course of progression to AML at month 7, one not related to mastocytosis at month 3 and three to rapid progression soon after 1 month of therapy. In three cases, midostaurin was used in the subsequent line of treatment (2 patients—2nd line, 1 patient—5th line).

Treatment is ongoing in 7 patients. The median follow-up period for the study group is 19 months (1–71 months). Detailed characteristics of patients who responded to treatment are presented in Table 2.

### 3.4. Safety profile

The most common nonhematologic adverse events were nausea (n = 6. 46%), vomiting (*n* = 5. 38%) and diarrhea (*n* = 5. 38%), mostly in 2 grade of the severity. Symptoms were most severe during the first 10 days of treatment. Symptomatic medications (ondansetron, loperamide, fluids) brought clinical improvement in most cases. Two patients required chronic use of antiemetics throughout the course of midostaurin.

Four patients (31%) developed infections (grade 2 bronchitis, *n* = 1; grade 3/4 pneumonia, *n* = 3). In two of these patients with pre-existing neutropenia, grade 4 neutropenia was occurred. One patient additionally developed pulmonary embolism. Symptomatic treatment in accordance with the current guidelines (broad-spectrum antibiotic therapy, a granulocyte colony-stimulating factor (G-SCF) and anticoagulants) with an interruption in chemotherapy were administered. The improvement of clinical condition was achieved in two cases. Two patients died of pneumonia after 1 month of therapy.

Grade 4 neutropenia and anemia occurred respectively in 3 (23%) and 2 (15%) patients, all with preexisting cytopenias.

Secondary acute myeoloid leukemia (AML) developed in one patient during the 7th month of treatment. Patient died on the 3rd day of induction therapy.

In summary, treatment was temporary discontinued in five patients, in four due to adverse events. The dose of midostaurin was reduced to 50 mg twice daily in one case. Re-escalation to the initial dose was possible in two patients (15%).

Detailed data on adverse effects of midostaurin in the study group are presented in Table 3.

## 4. Discussion

ASM is an orphan disease with an estimated global prevalence of 1/250,000–400,000 [16]. The diagnosis of ASM is an indication for cytoreductive therapy [14]. The choice of therapy is a huge challenge for clinicians. So far, various treatment strategies have been used, including cladribine (2-CDA), interferon alfa (IFN-α), as well as classic cytostatics (hydroxycarbamide (HU), cytarabine or fludarabine). However, all of the options mentioned above did not bring satisfactory response rates, thus emphasizing the need to develop innovative therapies [17,18].

For many years, IFN-α was considered the first-line treatment in patients with all subtypes of advanced mastocytosis. However, previously published results of case reports or small trials (*n* ≤ 15) showed its inconsistent efficacy for the treatment of ASM. The exact dosage and duration of treatment remained uncertain. Dosage from 0.5 to 10 million units three times a week have been used. Overall response rate (ORR) of up to 60% was observed. The frequency of major remission (MR) was approximately 20–30%. The time to best response may be a year or longer and delayed responses to therapy have been described. IFN-α therapy have been provided control of MC mediator releases symptoms, resolution of urticaria pigmentosa (UP), improvement of skeletal disease, improvement in anemia and/or reduction in bone marrow MC burden. Nevertheless, a high rate (up to 50%) of IFN-α adverse effects have been recorded. The main adverse events were fatigue, flu-like symptoms, fever, bone pain, depression, and thrombocytopenia [19,20,21].

2-CdA has shown therapeutic activity in all subtypes of SM. However, it should be emphasized that when we narrow the analysis only to ASM, the number of patients treated with 2-CdA has been relatively small. The largest study on the efficacy of 2-CdA in mastocytosis to date has been reported by the French Group. Barete et al. enrolled 14 patients with ASM. Patients received median of 4 courses of 2-CdA. ORR of 43% was observed with the best response on MR level (36%). The efficacy was confirmed for MC mediator release symptoms as well as in infiltration-related symptoms especially for those involving skin, gut, and fatigue. However, no complete remission (CR) was reported. The most common adverse effects were leucopenia and opportunistic infections [22].

In a recently published Polish study, Helbig et al. presented a series of 7 ASM patients treated with 2-CdA. Authors presented similar results. All patients received 2-CdA intravenously for 5 consecutive days. Median dose per cycle was 45 mg (range 35–60). Median number of cycles was 6 (range 1–7). ORR was estimated at 66%. Median duration of response was 1.98 years (range 0.2–11.2). Clinical improvements were mainly observed in terms of hematological parameters, hepatosplenomegaly and mediators release symptoms. A decrease of serum tryptase levels was noted in all patients. However, mast cell infiltration in BM decreased after therapy by less than 50% when compared with baseline. Treatment was generally well tolerated. Infection, grade 2–4 neutropenia and grade 2 thrombocytopenia were reported [23].

Taking into account the effect of KIT mutations in the pathogenesis of SM, tyrosine kinase inhibitors (TKI) have become an attractive new direction of therapy (17). So far, only 2 TKI’s have been approved by the FDA for the treatment of patients with advanced SM including ASM: Imatinib and midostaurin [24,25].

Imatinib is effective in the presence of the wild-type *KIT* variant, *PDGFR* and BCR-ABL, but is not active against the *KIT D816V* mutation [26]. The *KIT D816V* mutation induces structural changes at the tyrosine kinase binding site, resulting in reduced affinity for type I TKIs such as imatinib, which recognize an active kinase conformation [27]. As a consequence, the drug fails to have an effect in SM patients with the KIT D816V mutation [28]. It is therefore of limited use in the treatment of rare cases of SM that show imatinib-sensitive *KIT* mutation (*F522C*, *K5091*, *V560G*, *V559G*, and *del419*) [29,30], without the *KIT D816V* mutation or with its unknown status [30,31]. This indication concerns only about 10% of SM patients. There are only limited clinical data from single case reports and small series of ASM cases available. No CR was reported. The percentage of the response to treatment is estimated at around 20–30%. The main adverse effects of treatment are diarrhea, edema and infections [27,32].

Midostaurin (PKC412) is the first multi-pathway TKI. It demonstrates the ability to inhibit signaling and cell proliferation pathways through the FLT3 receptor. It induces the mechanisms of apoptosis in leukemic cells, which show mutations in the *FLT3* gene with the tandem duplication type within the transmembrane domain (ITD) and mutations in the kinase domain (TKD). It has the ability to inhibit KIT signaling pathways (wild-type and *D816V* mutant), cell proliferation and histamine release, and induce apoptosis in MCs. In in vitro studies, it inhibits the activity of several other tyrosine kinases, such as PDGFRα/β, VEGFR2 and the serine/threonine PKC family kinases [26,33].

116 patients with advanced SM were enrolled in the midostaurin registration study #CPKC412D2201. Efficacy of treatment was assessed in 89 patients, including only 16 with pure ASM. The drug was administered orally at a dose of 100 mg twice daily. The median follow-up was 26 months (range 12–54). Overall ORR was estimated at 60%, with 45% achieving MR and 15% achieving PR. Median overall survival (OS) was 28.7 months and median progression-free survival (PFS) was 14.1 months. In the group of patients who responded to treatment, the median duration of response was 24.1 months and the median OS was 44.4 months, with the best response rate in ASM patients at 75%—mean duration of response was not achieved in this group. Responses occurred regardless of the *KIT D816V* mutation status. During treatment, resolution of hypoalbuminaemia (58%), red blood cell (40%) and platelet (100%) transfusion independence, improvement in liver function and/or weight gain (25%) was observed. Significant (>50%) reduction in bone marrow infiltration by MCs and a decrease in tryptase levels were observed. The most common adverse events were nausea, vomiting and diarrhea. Hematological adverse effects: Grade 3 or 4 neutropenia, anemia and thrombocytopenia occurred in 24%, 41%, and 29% of patients respectively, mainly those with pre-existing cytopenias [12].

De Angelo et al. presented data from phase II trial of midostaurin in advanced SM. Midostaurin was administered at 100 mg twice daily, as 28-day cycles in 26 patients. In the study group, there were only 3 patients diagnosed with ASM. One of them responded to treatment and achieved partial response (PR) [34].

Real-world data have a great value, especially in the case of rare diseases. At the Congress of The American Society of Hematology in 2019, Gajra et al. presented their observations on the real-world utilization of midostaurin in the treatment of patients with SM. In the group of 38 patients treated with midostaurin—33 were diagnosed with ASM. The median age at diagnosis was 63 years (range 27–79), with the majority of patients being females (61%). Midostaurin was used in the first line in 63%, in the second in 32%, in the third in 5% of patients. With a median follow-up of 13.4 months from the start of midostaurin treatment, the median duration of midostaurin was only 2.4 months (95% CI 1.1–4.6). The authors pointed to the potential barriers to the proper use of midostaurin in SM, such as lack of awareness, lack of access to the drug or delayed SM diagnosis due to its rarity [35].

In this paper, we present our own experience in the treatment of patients diagnosed with ASM who received midostaurin in the early access program. According to our knowledge, this is the first report on this topic in the literature.

The use of midostaurin brought clinical benefit in mostly (77%) our patients with ASM. We observed an improvement in general condition, relief of symptoms related to the release of mediators, weight gain or stabilization, improvement in blood count parameters, decreased need for red blood cells transfusion, reduction in liver and spleen enlargement, significant decrease in bone marrow infiltration by mast cells and a decrease in serum tryptase concentration.

Gastrointestinal disturbances, such as nausea, vomiting, and diarrhea related to the treatment, are common adverse effect of midostaurin. We recommend initiating therapy in a hospital setting, especially in the elderly or in patients with worse performance status. Importantly, symptomatic treatment, adequate hydration, and systematic use of prophylaxis with H1 and H2 antagonists bring clinical.

ASM is a highly heterogeneous disease. The obtained data, despite the fact that they concern a small group of patients, emphasize the problem, which is an enormous challenge in the treatment of patients with ASM in everyday clinical practice. In the study group, midostaurin did not provide any clinical response in three patients. In two of these, a very aggressive clinical course of ASM was observed, the patients died shortly after diagnosis. There is an unquestionable need to look for prognostic and predictive factors. One of the limitations of the study is the lack of molecular tests (*ASXL1*, *RUNX1*, *SRSF2*) which could allow to calculate one of known prognostic models (WHO or Mayo) [36,37].

Despite the progress made over the past two decades in diagnosing and improving the classification of various forms of MC disease, there are still many controversies regarding the eligibility of patients for cytoreductive therapy and the monitoring and assessment of response to treatment. The main cause of clinical dilemmas is the presence of non-measurable “C” findings of organ damage in the course of SM (osteolytic lesions in the bones, ascites, weight loss). There are many diagnostic uncertainties, organ dysfunction and cytopenias may be multifactorial (e.g., MC thrombocytopenia, hypersplenism, AHN, or bleeding), and symptoms of SM and AHN may overlap. There are no clear recommendations regarding the preferred methods for the diagnosis of organ changes (e.g., recommended tests assessing bone condition, gastrointestinal involvement, exudates). Finally, monitoring response to treatment remains a challenge. Hitherto published reports have used highly heterogeneous response criteria, e.g., response to treatment was defined by a reduced degree of bone marrow infiltration by MCs, or an improvement in clinical symptoms associated with the release of MC mediators. In 2013, specific response criteria for advanced mastocytosis were defined by the IWG-MRT-ECNM. These are the currently valid modified Valent criteria, which were used, among others, to evaluate response in the registration study of midostaurin [5].

Although there is no exact data as to what percentage of patients with advanced SM have unmeasurable “C” findings, it seems that this group may be significant. In #CPKC412D2201 study, out of 116 enrolled patients, in 14 “C” symptoms were not measurable and in 13 were considered unrelated to mastocytosis [12]. In our study, five patients (38%) had no measurable “C” symptoms and yet we diagnosed them with ASM. We considered the improvement of ASM symptoms, a decrease of serum tryptase concentration and decrease of bone marrow MC burden as sufficient indicators of clinical response to midostaurin in these subgroups of patients.

## 5. Conclusions

Our observations suggest potential efficacy and acceptable safety profile of midostaurin in patients with ASM. However, further studies on larger population of patients and with longer follow-up are necessary to evaluate its real value. It is difficult to compare the efficacy of midostaurin in ASM therapy with the previously used regimens. So far, head-to-head studies have been not performed. This highlights the importance of the ongoing efforts to increase the amount of real-world data on the treatment of ASM, in order to enhance physicians’ decision-making process.

## Figures and Tables

**Figure 1 jcm-10-01109-f001:**
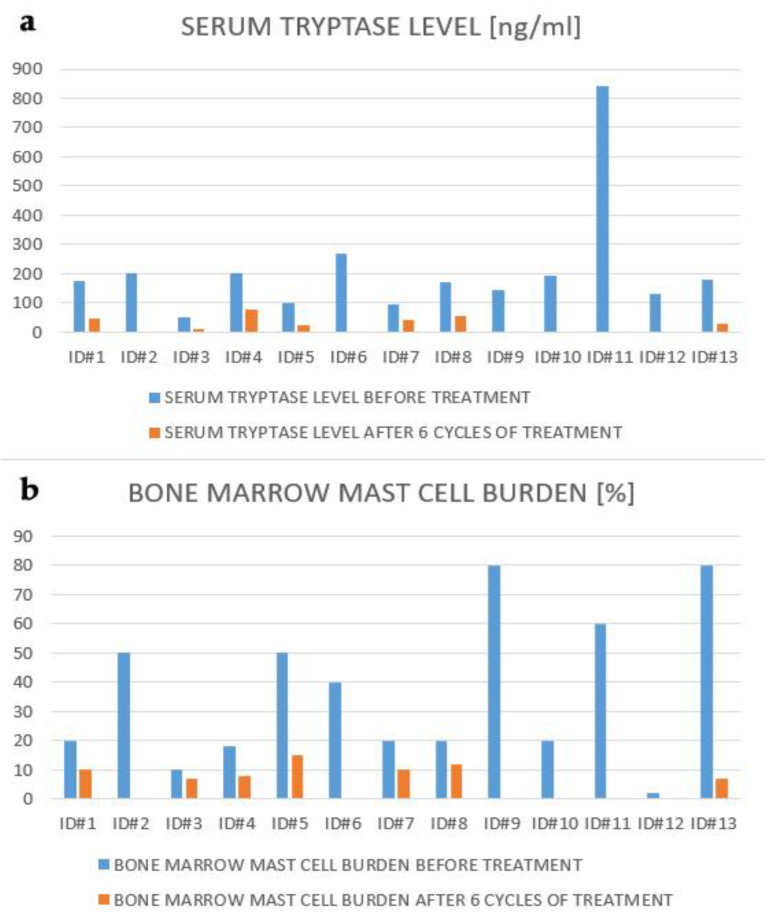
Clinicopathological measures of response: Serum tryptase level (**a**), bone marrow mast cell burden (**b**).

**Figure 2 jcm-10-01109-f002:**
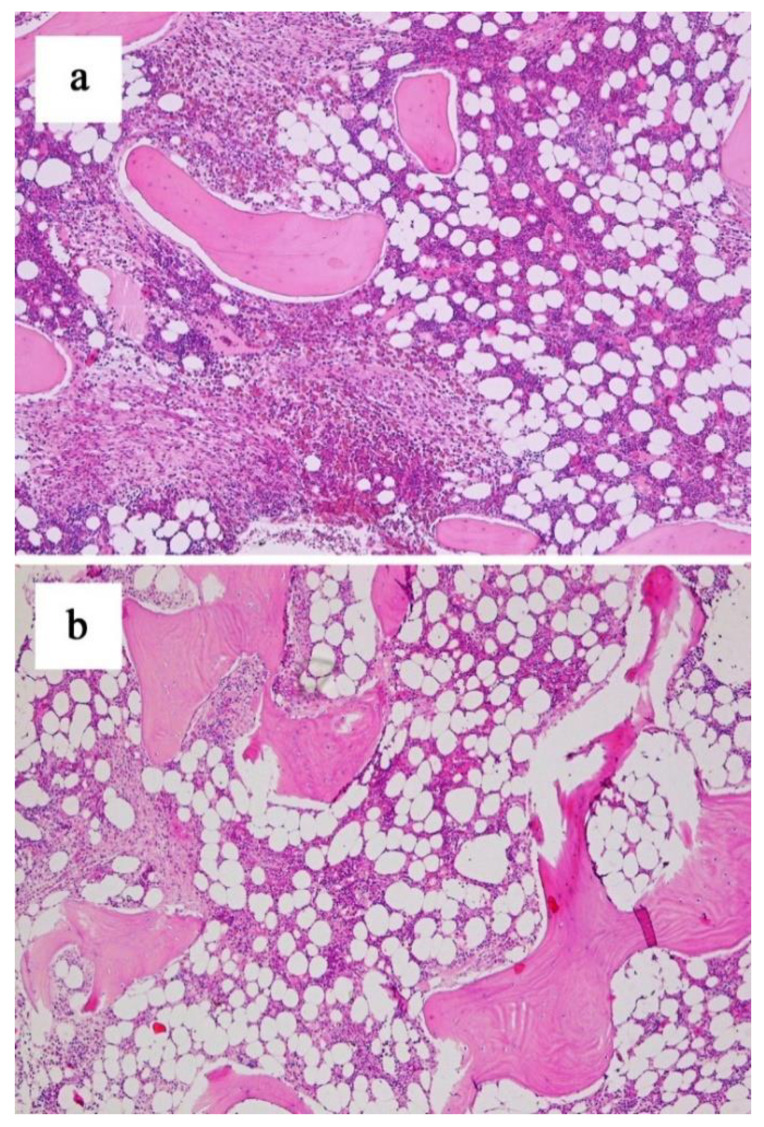
Bone marrow mast cell burden before (**a**) and after 6 cycles of treatment (**b**) (patient ID#4) (H + E; objective magnification ×4).

**Table 1 jcm-10-01109-t001:** Demographics and clinical characteristics of patients.

Characteristic	*N* (Value)
Male/female	5/8
Age—yrs (range)	63 (range: 48–79)
ECOG performance status	
0,1	5
2,3	8
Mutation status	
*KIT D816V* positive	11
*KIT D816V* negative	1
*KIT D816V, D816H* positive	1
Disease burden	
Splenomegaly	9
Hepatomegaly	9
Ascites	3
Lymphadenopathy	4
Bone lesions	12
Large osteolyses	8
Skin lesions	3
Grade 2 hypoalbuminemia	1
Transfusion dependent anaemia	2
Grade 2 anemia	5
Grade 4 neutropenia	2
Grade 3 thrombocytopenia	2
Grade 2 thrombocytopenia	1
Grade 1 thrombocytopenia	2
Serum tryptase level—ng/mL	174 (range: 49.9–841)
Bone marrow mast-cell burden—%	20 (range: 2–80)
No. of C-findings per patient—no. of patients	
1	4
2	2
3	4
4	1
5	2
Creatinine clearance, mL/min	
30–50	1
>50	12
Hemoglobin (HGB) (g/dL)	10.4 (range: 7.6–15.2)
Platelets (PLT) (K/uL)	156 (range: 40–307)
Eosinophils (EOS) (K/uL)	0.16 (range: 0–2.779)
Lactate dehydrogenase (LDH) (U/L)	239 (range: 112–324
Alkaline phosphatase (ALP) (U/l)	137 (range: 36–293.8)
No. of prior regimens for ASM	
0	5
1	6
4	2
Median time from start of first-line treatment to start of midostaurin therapy, months	6.5 (range: 4–62)
Prior regimens for ASM	
Cladribine	7
Imatinib	1
Interferon alfa	2
Hydroxycarbamide	1
Dasatinib	1
Midostaurin exposure (months)	9 (range: 1–21)

**Table 2 jcm-10-01109-t002:** Clinical diagnosis and outcomes in patients.

Patient no.	Sex	Age (Yrs)	ECOG	C-Findings	No. of C-Findings	Serum Tryptase Level [ng/mL]	Mutation Status	No. of Prior Regimens for ASM	Prior Regimens for ASM	Time from Start of First-Line Treatment to Start of Midostaurin Therapy	Bone marrow Mast-cell Burde*n*—%	Response	Total Cycles of Midostaurin	Survival Condition	Survival Time (Mths)
1	M	48	2	Neutropenia < 1 × 10^9^/L Hepatosplenomegaly with ascites Weight loss	3	>200	KIT D816V negative	1	Imatinib	14	50	MiR, PR	6	Die	21
2	F	60	1	Bone lesions	1	174	KIT D816V positive	1	Cladribine	7	20	Not evaluated *	21	Survive	31
3	M	52	1	Bone lesions	1	49.9	KIT D816V positive	0	NA	NA	10	Not evaluated *	10	Survive	21
4	F	56	1	Anaemia Bone lesions	2	>200	KIT D816V positive	0	NA	NA	18	InR, MR	20	Survive	21
5	M	67	2	Bone lesions Weight loss	2	99.8	KIT D816H and D816V positive	1	Cladribine	6	50	Not evaluated *	19	Survive	25
6	F	68	3	Anemia Hepatosplenomegaly with ascites Weight loss Hypoalbuminemia	4	268	KIT D816V positive	4	Cladribine, Interferon alfa, Hydroxycarbamid, Cladribine	16	40	Not evaluated †	3	Die	19
7	F	70	2	Bone lesions	1	95.5	KIT D816V positive	1	Cladribine	6	20	Not evaluated *	12	Survive	19
8	F	63	1	Bone lesions	1	171	KIT D816V positive	0	NA	NA	20	Not evaluated *	13	Survive	13
9	F	61	1	Anemia Thrombocytopenia Hepatomegaly with impaired liver function	3	142	KIT D816V positive	1	Cladribine	5	80	PR	9	Survive	15
10	M	79	2	Anemia Thrombocytopenia Hepatomegaly with impaired liver function Bone lesions Weight loss	5	196	KIT D816V positive	1	Cladribine	4	20	PD	2	Die	6
11	F	66	3	Anemia, Bone lesions, Weight loss	3	841	KIT D816V positive	0	NA	NA	60	PD	1	Die	2
12	M	71	3	Anemia Neutropenia Thrombocytopenia Hepatomegaly with impaired liver function Weight loss	5	131	KIT D816V positive	0	NA	NA	2	PD	1	Die	1
13	F	61	0	Anemia Thrombocytopenia Weight loss	3	180	KIT D816V positive	4	Cladribine, Interferon alfa, Cladribine, Dasatinib	62	80	PR	7	Survive	71

M male, F female, ECOG Eastern Cooperative Oncology Group, BM bone marrow, MiR minor response, MR major remission, PR partial response, InR incomplete remission, NA not applicable, NT not done, PD progressive disease, PR partial remission. Reasons that patients could not be evaluated for response were: † not enough time receiving treatment and death (1 patients). * absence of measurable C-findings (5 patients).

**Table 3 jcm-10-01109-t003:** Adverse events in the patients.

Event	Any Grade (*n*,%)	Grade 3 or 4 (*n*, %)
Nonhematologic adverse events		
Nausea	5 (38%)	1 (7%)
Vomiting	4 (31%)	1 (7%)
Diarrhea	4 (31%)	1 (7%)
Fatigue	3 (23%)	0
Pneumonia	3 (23%)	3 (23%)
Bronchitis	1 (7%)	0
Hypertransaminesemia	1 (7%)	0
Hematologic abnormalities		
Neutropenia	3 (23%)	3 (23%)
Anemia	2 (15%)	2 (15%)

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
