# Peer review of "Real-World Efficacy of Midostaurin in Aggressive Systemic Mastocytosis"

_jcm, 2021, doi:10.3390/jcm10051109_

Round 1
Reviewer 1 Report
Szudy-Szczyrek et al are presenting a real-world data for ASM. Single center experience with 8 patients treated after the FDA approval for midostaurin. There are 2 phase II trials with larger number of patients treated in a controlled sitting to assess this agent for ASM. I do believe that real-world data is an important aspect of cancer treatment outcomes as sometimes results of clinical trials cannot be reproduced in general hematology/oncology practices.
The limiting factor to this study is the number of patients. One possibility the authors can consider is comparing this cohort to patients treated before 2017 in the same center with older treatment (INF, 2-cda, UH, etc..), not to compare survival or response rates as we already know those older treatments have limited activity but to examine aspects of midostaurin as this agent has side effects require expertise to manage leading to shorter duration of treatment.
I might not agree with the authors conclusion in the abstract for “high efficacy” as there is unmet need for better and tolerated treatment in ASM specially to increase the duration of therapy as it might lead to higher response rate. The ORR of midostaurin is 60s% and would consider highly effective treatment when we see high CR rate.
Additionally, would like to see if the authors have performed more molecular testing (ASXL1, RUNX1, SRSF2) to calculate one of the prognostic models (WHO or Mayo).
Author Response
Dear Reviewer,
We appreciate the time and effort put into the review and are grateful for the valuable comments which allowed us to improve the manuscript. We have incorporated the suggested changes and marked them in the manuscript by using bold font and yellow highlighting.
Below, in blue, we have responded to your comments:
The limiting factor to this study is the number of patients. One possibility the authors can consider is comparing this cohort to patients treated before 2017 in the same center with older treatment (INF, 2-cda, UH, etc..), not to compare survival or response rates as we already know those older treatments have limited activity but to examine aspects of midostaurin as this agent has side effects require expertise to manage leading to shorter duration of treatment.
> We have contacted other centers in Poland that treat agressive systemic mastocytosis (ASM) and expanded the study group by 5 patients. We focused on ASM, so deal with a limited number of patients. Midostaurin registration study #CPKC412D2201 included 116 patients, however only 16 of them suffered from ASM. Previosuly published data on ASM is mostly based on case report series. We discussed issues related to midostaurin treatment in comparison with previously available options. Unfortunately, we have not kept any mastocytosis registry so far. In the Discussion Section we have additionaly cited one recently published Polish study [Helbig et al.].
I might not agree with the authors conclusion in the abstract for “high efficacy” as there is unmet need for better and tolerated treatment in ASM specially to increase the duration of therapy as it might lead to higher response rate. The ORR of midostaurin is 60s% and would consider highly effective treatment when we see high CR rate.
> We do agree with the Reviewer’s comment. We have modified the manuscript and removed the conclusion regarding the “high efficacy” .
Additionally, would like to see if the authors have performed more molecular testing (ASXL1, RUNX1, SRSF2) to calculate one of the prognostic models (WHO or Mayo).
> Unfortunatelly we have not performed ASXL1, RUNX1 or SRSF2 molecular tests. This is one of the limitations of the study, and was mentioned in the text.
Reviewer 2 Report
The paper concerns a current topic in mastocytosis, i.e. how to judge the response in unmeasurable organ damage. Nevertheless, the authors do not suggest a pratical approach to address this topic, and they report on a very small group of patients.
Author Response
Dear Reviewer,
Thank you for the time and effort put into the review. We are grateful for your input, which was extremly valuable and allowed us to improve the quality of the manuscript. We have performed changes in the text and marked them by using bold font and yellow highlighting.
Below, in blue, we have responded to your comments:
The paper concerns a current topic in mastocytosis, i.e. how to judge the response in unmeasurable organ damage. Nevertheless, the authors do not suggest a pratical approach to address this topic, and they report on a very small group of patients.
> Among patients with unmeasurable "C" changes, we considered the improvement of general condition, a decrease of serum tryptase concentration and decrease of bone marrow mast cell burden as sufficient indicators of clinical response to midostaurin. It was mentioned in the text.
To expand the study group we have contacted other centers in Poland that treat agressive systemic mastocytosis (ASM) which allowed us to include 5 additional individuals. We focused on ASM, so deal with a limited number of patients. Midostaurin registration study #CPKC412D2201 included 116 patients, however only 16 of them suffered from ASM. Previosuly published data on ASM is mostly based on case report series.
Round 2
Reviewer 2 Report
The effort to improve the number of cases and the hystorical comparison have brought an improvement of quality of paper.